

# An update on the RTTOV fast radiative transfer model (currently at version 12)

Roger Saunders[1], James Hocking[1], Emma Turner[1], Peter Rayer[1], David Rundle[1], Pascal Brunel[2], Jerome Vidot[2], Pascale Roquet[2], Marco Matricardi[3], Alan Geer[3], Niels Bormann[3] and Cristina Lupu[3]

[1]Met Office, Fitzroy Rd., Exeter, U.K.
[2]MeteoFrance/CMS, Avenue de Lorraine, Lannion, France
[3]ECMWF, Shinfield Park, Reading, U.K.

*Correspondence to*: Roger Saunders (roger.saunders@metoffice.gov.uk)

**Abstract.** This paper gives an update of the RTTOV (Radiative Transfer for TOVS) fast radiative transfer model which is widely used in the satellite retrieval and data assimilation communities. RTTOV is a fast radiative transfer model for simulating top of atmosphere radiances from passive visible, infrared and microwave downward-viewing satellite radiometers. In addition to the forward model, it also optionally computes the tangent linear, adjoint and Jacobian matrix providing changes in radiances for profile variable perturbations assuming a linear relationship about a given atmospheric state. This makes it a useful tool for developing physical retrievals from satellite radiances, for direct radiance assimilation in NWP models, for simulating future instruments and for training or teaching with a graphical user interface. An overview of the RTTOV model is given highlighting the updates and increased capability of the latest versions and gives some examples of its current performance when compared with more accurate line by line radiative transfer models and a few selected observations. The improvement over the original version of the model released in 1999 is demonstrated.

## 1 Introduction

Over the past two decades fast radiative transfer models have become an indispensable tool for a variety of applications including data assimilation in numerical weather prediction (NWP) models (Eyre et al., 1993), enabling physical retrievals from satellite data (Li et al., 2000), producing simulated imagery from NWP models (Blackmore, 2014; Lupu and Wilhelmsson, 2016) and for assessing the performance of proposed instruments to fly on future satellites (Andrey-Andres et al., 2017). The RTTOV (Radiative Transfer for TOVS) model was developed to enable the direct assimilation of radiances during the 1990s at the European Centre for Medium-Range Weather Forecasts (ECMWF) when it was implemented within their variational system (Andersson et al., 1998). The development of RTTOV was subsequently taken on within the EUMETSAT-funded Numerical Weather Prediction Satellite Application Facility (NWP SAF) in 1998. There are over 1000 users worldwide of RTTOV and it is now used in many NWP centres around the world as part of their data assimilation system both for weather forecasting and producing atmospheric reanalyses. Although initially developed for the old TIROS Operational Vertical Sounder (TOVS) radiometers, RTTOV can now simulate around 90 different satellite sensors measuring in the microwave (MW), infrared (IR) and visible (VIS) regions of the spectrum. Some of these instruments flew in the 1970s and now RTTOV enables their radiances to be assimilated into historical atmospheric reanalyses exploiting these data, for the first time, with modern data assimilation methods (Poli et



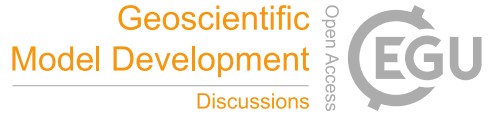

al., 2017). RTTOV is also part of the CFMIP Observation Simulator Package (COSP) (Bodas-Salcedo et al., 2011) used in climate model simulations to provide top of atmosphere radiances for climate model runs.

There have been several other fast models developed over the years notably the Community Radiative Transfer Model (CRTM) (Chen et al., 2008; Ding et al., 2011) which is also used in several NWP models, the Optimal
Spectral Sampling (OSS) model (Moncet et al., 2015), and many others which have taken part in several inter-comparisons with RTTOV (e.g. Garand et al., 2001; Saunders et al., 2007; Aumann et al., 2018).
These fast models are not only forward models (i.e. compute the top of atmosphere radiance from a given atmospheric state) but also compute the Jacobian matrix which gives the change in radiance for a change in any element of the atmospheric state assuming a linear relationship about a given atmospheric state. Not all
applications require the full Jacobian matrix to be stored and so tangent-linear and adjoint versions of the code are also provided as options. The performance of these models is not only assessed for the forward calculations but also for the Jacobian computations in terms of speed and accuracy (e.g. Saunders et al., 2007).

The initial version of RTTOV developed at ECMWF (version 3) which was made available to the community
was documented in the open literature (Saunders et al., 1999) but since then there have been many upgrades (it is now at version 12 described in detail by Saunders et al. (2017)) and so this paper is intended to provide an updated overview description of RTTOV in the peer review literature taking into account all the changes in the interim period. The full documentation of the latest supported versions of RTTOV is available from the NWP SAF web site[1] and there have been papers on various aspects of the RTTOV development which are referenced
here for more details. Section 2 provides a brief history of the different versions of RTTOV, section 3 describes the latest capabilities of the model at version 12, section 4 shows how well it reproduces the line-by-line models on which it is trained and section 5 makes some comparisons with observations. A summary and future plans are given in section 6.

## 2   A brief history of RTTOV

With the advent of satellite sounding radiometers in the 1970s and the need to derive atmospheric profile retrievals efficiently in near real time, activities were initiated to develop fast radiative transfer models for this application. An initial study reported by McMillin and Fleming (1976) showed that the layer transmittance of the atmosphere, for a region with only well mixed gaseous absorption, can be parameterised by functions of the mean layer temperature. Further work developed formulations for water vapour and ozone where the gas
concentration was also taken into account (Eyre and Woolf, 1988) and gradient versions of the model were developed for profile retrievals and assimilation applications which culminated in the first version of the RTTOV model maintained at ECMWF in the early 1990s (Eyre, 1991). During the mid 1990s EUMETSAT were setting up their SAFs one of which was the NWP SAF led by the Met Office (UK) which aimed to provide software packages to enable NWP centres to better exploit satellite data in their NWP systems. RTTOV was adopted by
the NWP SAF as one of its main packages and has been developed within the SAF ever since and distributed to users worldwide with currently over 1000 users of RTTOV in 2017.

---

[1] http://nwpsaf.eu

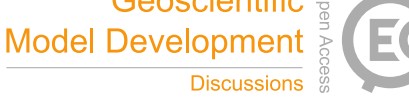



The ECMWF version of RTTOV in 1999 was described in Saunders et al. (1999). Since then there have been many enhancements developed under the NWP SAF activities and the interested reader is referred to the RTTOV web site and the various science and validation reports for full details. For example one new innovation in recent years has been to compute the infrared spectrum as principal component scores, PCs (e.g. PCRTM

(Liu, 2006), HT-FRTC (Havemann et al., 2009), and PC-RTTOV (Matricardi, 2010)). PC-RTTOV and HT-FRTC have been adopted as options within RTTOV and this has enabled experiments to assimilate the PCs directly in NWP systems (Matricardi and McNally, 2014). This is potentially a way to make use of more of the spectrum measured by the new advanced IR sounders such as the Infrared Atmospheric Sounding Interferometer (IASI) as traditional methods can only assimilate a few hundred spectral channels efficiently with fast models.

Another major development has been the addition of scattering effects for simulating cloudy and aerosol-affected radiances in the IR in version 9 (Matricardi, 2005) and a wrapper code for computing scattering from hydrometeors at MW frequencies introduced in version 8 (Bauer et al., 2006).

Table 1 gives a summary of the major upgrades for each version of the model culminating in version 12 released

to users in February 2017. In addition to these each version benefitted from improved transmittances computed from the latest line-by-line models for the IR and MW wavelength regions (i.e. LBLRTM (Gordon et al., 2017, Rothman et al., 2013) or AMSUTRAN (Liebe et al., 1989, Saunders et al., 2017) and also improved computational speed through continuous optimisation of the code for parallel computing architecture. After every new version of the code was developed an extensive validation campaign was undertaken to ensure the

code was not slower or less accurate (when compared with line by line models) than the previous version. Checks were also made to ensure the tangent-linear, adjoint and full Jacobian versions of the code were all consistent with each other. Another constraint was that the code had to be backward compatible so users could reproduce the results of the previous version with the new code to enable a controlled transition to the new model in their operational systems.

| RTTOV version | Release date | Major enhancements |
|---|---|---|
| 1-4 | Mid1990's | • TOVS only on 40 atmospheric levels. Clear sky and black cloud. Surface emissivity provided by user. |
| 5 | 1999 (pre-NWPSAF) | • ATOVS, METEOSAT, GOES imagers. Clear sky and grey cloud, 43 atmospheric layers |
| 6 | Mar 2000 | • Revised water vapour transmittance calculation<br>• More sensors supported (e.g. ATSR, GOES, AVHRR, MODIS, GMS, SSM/I)<br>• Addition of SSIREM IR and FASTEM MW ocean surface emissivity models<br>• Addition of MW cloud liquid water absorption |
| 7 | Jan 2002 | • New clear air transmittance formulation introduced<br>• Improved cloud simulations for multi-layers<br>• FASTEM version 2 introduced |
| 8 | Nov 2005 | • Revised transmittance calculations for more variable gases and separate continuum<br>• FASTEM version 3 introduced to allow simulation of polarimetric radiometers<br>• Addition of RTTOV-SCATT wrapper for MW scattering from hydrometeors |
| 9 | Mar 2008 | • Addition of reflected solar radiation for SWIR Channels<br>• IR cloud and aerosol scattering added using parameterisation from Chou et al. (1999) and maximum-random cloud overlap<br>• Radiative transfer computation possible on user input pressure levels<br>• Coefficient files for advanced IR sounders provided on 100 levels.<br>• Internal profile interpolation added. |
| 10 | Jan 2011 | • First land surface emissivity atlases UWIREMIS and TELSEM and CNRM (Karbou |



| | | |
|---|---|---|
| | | et al., 2006). |
| | | • Introduction of FASTEM versions 4 and (later) 5 |
| | | • Computation of Principal Components for advanced IR sounders added (PC-RTTOV) |
| | | • Number of atmospheric layers increased from 43 to 51 for radiometers |
| | | • Included Zeeman effect for high peaking channels |
| 11 | May 2013 | • Ability to simulate VIS/NIR radiances for clear sky and basic cloud |
| | | • Land surface BRDF atlas |
| | | • FASTEM version 6 introduced |
| | | • Number of atmospheric layers increased from 51 to 54 for radiometers |
| | | • Improved profile interpolation options |
| | | • NLTE parameterisation introduced for shortwave IR channels |
| | | • RTTOV graphical user interface (GUI) created |
| 12 | Feb 2017 | • VIS/NIR/IR scattering using Discrete Ordinates. |
| | | • Developments to existing microwave emissivity atlases |
| | | • New IR sea surface emissivity model |
| | | • New IR land surface emissivity atlas (CAMEL) |
| | | • Added $SO_2$ as a variable gas |
| | | • First version of another PC model HT-FRTC included as an option |
| | | • NLTE parameterisation updated for shortwave IR channels |

Table 1. Major enhancements to RTTOV since the initial versions developed at ECMWF in mid 1990's.

Initially RTTOV only supported the NOAA TOVS radiometers (HIRS, MSU) but with the number of sounding radiometers increasing as more nations launched instruments and as TOVS was upgraded to the Advanced TOVS (ATOVS) the demand for RTTOV to simulate different satellite radiometers grew. Also with the recent

extension of RTTOV to cover the VIS and near IR parts of the spectrum more radiometers can be simulated which cover this region of the spectrum.

The list of satellites now supported by RTTOV v12 is up to 50, and increasing, and the full list of instruments, up to 90 currently supported is given on the RTTOV web site and latest RTTOV user guide (Hocking et al.,

2017). Users can request any satellite nadir-viewing radiometer be supported by RTTOV as long as the channel spectral response functions are provided. Many of the instruments are now retired, but they provide measurements since 1969 and are required in support of the global atmospheric reanalysis efforts which are under way. This has enabled satellite data to be used for climate monitoring applications.

## 3 Model formulation

The details of the formulation of the RTTOV model are documented in various reports over the past 20 years and it is not possible to reproduce all aspects of the model here. However an overview is given here with references for the more detailed aspects given where necessary. The main framework of the formulation is given in Saunders et al. (1999) and hasn't changed in the latest versions of RTTOV although there are many additional capabilities added.

### 3.1 Atmospheric profile and surface variables

### 3.1.1 Input state vector

The classical temperature and water vapour profiles are the default input profiles to RTTOV (ozone is an optional variable gas for infrared sensors and can be included as a mixed gas for the microwave sensors) but in addition the capability to simulate the transmittance from several atmospheric variable gases has been added.

RTTOV can also include in the state vector concentration profiles of ozone, carbon dioxide, nitrous oxide, methane, carbon monoxide and sulphur dioxide to compute their atmospheric transmittances. Cloud liquid/ice





water profiles and aerosol profiles can also optionally be provided to enable absorption/scattering calculations at IR and VIS wavelengths and cloud liquid water absorption to be computed for MW frequencies. The cloud fraction profile can also be provided to enable simulations for atmospheres partially covered by clouds. In that case the radiative transfer is solved by using the maximum random overlap method. The profiles can be input on

any user-defined pressure levels and these input profiles are then interpolated to the levels on which the RTTOV coefficients are supplied to compute the gaseous transmittance. The vertical layering for the coefficients has been optimised from the 40 layers used in the original version of RTTOV to 53 layers from 1050 hPa to 0.005 hPa for multi-channel radiometers with fairly wide spectral response functions and for all MW radiometers. These are defined in Table 1 of Hocking et al. (2017). For the new IR hyperspectral sounders (e.g. IASI on

Metop) coefficients on 100 layers (1100 hPa to 0.005 hPa) are the optimal configuration provided to users although 53 layer coefficient files are also available to reduce the run time of the model at the expense of accuracy. Once the gas optical depth profiles have been computed (see section 3.2) they are interpolated back to the user levels for the radiative transfer computation (see section 3.3) which is more accurate.

For the surface variables skin temperature, 2 m temperature and water vapour concentration, wind speed (over ocean only), surface type and elevation all have to be defined. To account for viewing angle effects the satellite zenith and optionally azimuth angles (for MW and VIS/NIR channels) at the surface are required. The nadir scan angle is also computed internally for the MW instruments where the mixing of the polarisations is a function of scan angle. The solar zenith and azimuth angles are also required if solar-affected simulations are required. The

surface emissivity/reflectance can be either input by the user or RTTOV can calculate it, for instance over the ocean using physical models such as ISEM (Sherlock and Saunders, 2000) or IREMIS (Saunders et al., 2017) for IR emissivities.  A sea surface solar BRDF model (Matricardi, 2003) is used for solar-affected channels, and FASTEM or TESSEM2 models (Prigent et al., 2017) at MW frequencies. The FASTEM model has had several updates during the development of RTTOV as the parameterisation has been improved to be valid for a wider

range of frequencies (Liu et al., 2011; Bormann et al., 2012; Kazumori and English, 2015). MW radiometers measure radiances that are usually a mix of vertically and horizontally polarised radiation, and so the effect of the reflecting surface on the different polarisations has to be taken into account before combining the radiances to compute a top of atmosphere value. For polarimetric radiometers such as Windsat all four components of the Stokes vector have to be taken into account when computing the total radiance measured by the radiometer.

Reflectance/emissivity atlases are provided over the land for visible and near infrared wavelengths (Vidot and Borbas, 2014; Vidot et al., 2018), for infrared UWIREMIS (Borbas and Ruston, 2010) and CAMEL (Borbas et al., 2017) and for the microwave TELSEM (Aires et al., 2011) and the CNRM atlas (Karbou et al., 2006; Karbou et al., 2010) which are all provided as part of the RTTOV package.

### 3.1.2 Profile training datasets

In order to compute the regression coefficients for RTTOV layer-to-space transmittances computed from line-by-line radiative transfer models using a diverse set of atmospheric profiles are used. For the visible and infrared transmittances they are stored in a database, whereas for the microwave they are produced at run time. The version of RTTOV in 1999 (Saunders et al., 1999) was trained on a dataset of 32 radiosonde profiles with 40 levels (Chevallier et al., 2000). Now diverse temperature, water vapour and ozone profiles which are

thermodynamically consistent are sampled from the ECMWF reanalysis fields (Chevalier et al., 2006) and for



the variable trace gas profiles the Copernicus Atmosphere Model reanalysis fields were also used (http://www.copernicus-atmosphere.eu/). Currently 83 atmospheric profiles on 101 or 54 levels are used to compute the layer to space transmittances (Matricardi, 2008).

The line-by-line model calculations are performed with some minor constituents that do not vary with profiles; these are called 'fixed gases'. The fixed gases included for RTTOVv11 are $O_2$, NO, $NO_2$, $HNO_3$, OCS, $N_2$, $CCL_4$, CFC-11, CFC-12, CFC-14. In RTTOVv12, $NH_3$, OH, HF, HCl, HBr, HI, ClO, $H_2CO$, HOCl, HCN, $CH_3Cl$, $H_2O_2$, $C_2H_2$, $C_2H_6$, and $PH_3$ were added. The profile concentrations are from the US Standard Atmosphere (1976). The new variable gas profile sets have been constructed to cover the variability observed

since the 1970's taking into account the fact that the mean profile should also be representative of the current state of the atmosphere. It is worth being aware of which year the $CO_2$ profile used for the transmittance calculations is valid for, as it has changed significantly during the satellite era. Similar considerations apply to the $CH_4$ and CFCs profiles assumed in the coefficient generation. More details on the latest variable gas profile datasets can be found in the RTTOVv12 science and validation report (Saunders et al., 2017).

**3.2 Transmittance model**

The physical basis of the fast model to compute atmospheric transmittance has not changed much since the original ideas of McMillin and Fleming (1976) and Eyre and Woolf (1988) where the layer optical depth for a specific gas and channel is parameterised in terms of layer mean temperature, absorber amount, pressure and viewing angle which are predictors for the optical depth for layer $j$ and $\sigma_j$ is the level $j$ to space optical depth for

that gas using the following formulation:

$$\sigma_j = \sigma_{j-1} + \sum_{k=1}^{k=m} a_{j-1,k} X_{j-1,k} \quad j = 2 \; to \; n \tag{1}$$

where $j$ is the level number where there are $n$ levels, $m$ is the number of predictors indexed by $k$ and $X_{j,k}$ are the predictors and $a$ are the coefficients for $n$ levels and $k$ predictors. The diverse profile datasets are used to compute layer optical depths for each gas and combinations of gases from a line by line model. The

LBLRTMv12.2 model (Clough et al., 2005) with AER v3.2 molecular database and MT-CKD2.5.2 for continuum absorption is used to calculate the VIS/near IR layer optical depths in the range 2000-25500 $cm^{-1}$ at a spectral resolution of 0.01$cm^{-1}$ and in the IR (175-3300$cm^{-1}$) at 0.001$cm^{-1}$ on 54 and 101 levels. The AMSUTRAN model, the 1989 version of the Liebe Millimeter-wave Propagation Model (MPM-89) (Liebe et al.,1989) is used for microwave instruments at all frequencies below 1000 GHz with a parameterisation to

include the Zeeman effect for high peaking channels around the 50-60 GHz oxygen lines. Spectroscopic parameters have been updated for the 22 GHz and 183 GHz water vapour lines based on half-width data from Liljegran et al. (2005) and Payne et al. (2008), respectively. All oxygen line parameters are updated to those from Tretyakov et al. (2005), and 35 ozone lines below 300 GHz from the HITRAN molecular database have been added. The strategy has been to update the reference line-by-line model calculations at least once every 5

years to benefit from improved spectroscopic databases and diverse profiles. The transmittance calculations are computed for all the diverse set of atmospheric profiles resulting in a large database of level to space transmittances and associated profile variables used for the statistical regression and hence coefficient generation. In the fast model the optical depths for mixed gases and each variable gas are computed from equation 1, converted into transmittances and then combined into 'effective' transmittances as ratios according to

the following formulation originally recommended by McMillin et al. (1995):





$$\tau_{i,j}^{tot} = \tau_{i,j}^{mix} \cdot \frac{\tau_{i,j}^{mix+wv}}{\tau_{i,j}^{mix}} \cdot \frac{\tau_{i,j}^{mix+wv+oz}}{\tau_{i,j}^{mix+wv}} \qquad (2)$$

This ratioing of transmittances can prove cumbersome when adding more variable gases and a more recent paper by McMillin et al. (2006) suggests a simpler approach may be feasible but this has not been implemented in RTTOV to date. The layer optical depths for mixed gases and each variable gas are combined to give the total

layer transmittance as in equation 2.

Over the years there has been research on improving the predictors used and there are now three possible sets of predictors, $X_{j,k}$, which can be invoked when running RTTOV:

- The original predictors: $H_2O$ and $O_3$ variable, all other gases fixed (referred to as v7 in RTTOV guide)

- Updated predictors that include $CO_2$ as a variable gas (referred to as v8 in RTTOV guide)

- Updated predictor set (referred to as v9 in RTTOV guide) optimised for water vapour channels, allow for additional optional variable trace gases ($N_2O$, CO, $CO_2$ and $CH_4$) and are designed to enable inclusion of solar radiation and zenith angles beyond 60 deg.

The original v7 predictors defined in Table 2 of Saunders et al. (1999) and the profile variables in Table 3 predicts the classical mixed gas, water vapour and ozone absorption and is still optimal for radiometers such as HIRS and AMSU. The v8 predictors described in Matricardi (2003) separates out the water vapour continuum from the line absorption and includes carbon dioxide as an additional variable gas which can be useful for historical instruments allowing for the increasing carbon dioxide. The most recent v9 predictor set described in

Matricardi (2008) is focused on getting the best optical depths from the hyperspectral IR sounder channels and can optionally include $CO_2$, $N_2O$, CO, $CH_4$ and more recently $SO_2$ as variable gases. It also extends the range of zenith angles the regression is valid for in the shortwave IR. Here there is a complex mix of transmittance ratios used depending on which spectral region being predicted and which is the dominant absorbing gas and so there are many different combinations of variable gas transmittance ratios used in equation 2. This predictor set is the

only one used for solar-affected radiances due to the large range of zenith angles the ray path can traverse in this case.

### 3.3 Radiative transfer

The radiative transfer calculation in RTTOV is now performed on the user-defined pressure levels input to RTTOV which is a change from the original version where the input layers and the levels on which the

transmittance were computed were the same. This allows more accurate calculations for cloud-affected radiances as the cloud top can be defined by the user at the required level. To enable this the input/output profiles must be interpolated from/to user levels to/from the levels on which the RTTOV coefficients are provided (normally 54 or 101). There are various options for the interpolation which can take as input a fixed vertical pressure grid or a variable pressure grid to allow 'sigma' coordinates commonly used in NWP models to be used. A tangent linear

and adjoint of the interpolation scheme are also included with one option described by Rochon et al., (2007). A description of the interpolation options used in RTTOV is given in Hocking (2014) and its application in the ECMWF model in Lupu and Geer (2015).





A radiative transfer model for simulating top of atmosphere satellite radiances has to compute the following radiative transfer equation:

$$L(v, \theta_{sat}, \theta_{sun}) = (1 - N)L^{Clr}(v, \theta_{sat}, \theta_{sun}) + NL^{Cld}(v, \theta_{sat}, \theta_{sun}) \tag{3}$$

where $L^{Clr}(v, \theta_{sat}, \theta_{sun})$ and $L^{Cld}(v, \theta_{sat}, \theta_{sun})$ are the clear sky and overcast sky radiances at a frequency v and zenith angle $\theta_{sat}$ and solar zenith $\theta_{sun}$. $N$ is the effective fractional cloud amount (i.e. the product of the fractional cloud amount and the cloud emissivity assuming it is grey body). The top of atmosphere clear sky radiance includes the emitted radiation from the surface and reflected downward radiation (emitted, solar and diffuse) and the emitted radiation from the atmosphere:

$$L^{Clr}(v, \theta_{sat}, \theta_{sun}) = \tau_s(v, \theta). \epsilon_s(v, \theta). B(v, T_s) + \int_{\tau_s}^1 B(v, T)d\tau + \left(1 - \epsilon_s(v, \theta_{sat})\right)\tau_s^2(v, \theta_{sat}) \int_{\tau_s}^1 \frac{B(v,T)}{\tau^2}d\tau +$$

$$L_{Sol}(v, \theta_{sat}, \theta_{sun}) \tag{4}$$

where $\tau_s$ is the surface to space transmittance, $\epsilon_s$ is the surface emissivity and $B(v, T_s)$ is the Planck function for the defined frequency and skin temperature. $L_{Sol}(v, \theta_{sat}, \theta_{sun})$ is the direct and diffuse solar radiation reflected

from the surface given by:

$$L_{Sol}(v, \theta_{sat}, \theta_{sun}) = \int_{\tau_s}^1 J^\uparrow(v, \theta_{sat}, \theta_{sun})d\tau + r_s(v, \theta_{sat})\tau_s^2 \int_{\tau_s}^1 \frac{J^\downarrow(v, \theta_{sat}, \theta_{sun})}{\tau^2}d\tau \tag{5}$$

where $J^\uparrow$ is the upwelling source function, $r_s(v, \theta_{sat})$ is the surface reflectance for the downward incoming radiance and upward outgoing radiance along the satellite line-of-sight. In fact this value is not available within

RTTOV so the input BRDF for the incoming solar and outgoing satellite surface zenith angles (multiplied by the cosine of the satellite zenith angle) is used instead. In general this should not cause significant errors since the surface-reflected downwelling radiation is very much smaller in magnitude than the upward-scattered component except for highly reflective surfaces. The upwelling and downwelling contributions are calculated for each layer of the input user level profile. The solar and satellite angles (and hence the phase function) are

assumed constant through each layer. A value for the source term for atmospheric layer $i$ (bounded by levels $i$ and $i+1$) is obtained by integrating over the layer:

$$J_i^{\uparrow\downarrow}(v, \theta_{sat}, \theta_{sun}) = F_{sun}\tau_{sun,i} \frac{P(\theta_i^{\uparrow\downarrow})}{4\pi} \frac{\sigma_s}{\cos(\theta_{sat,i})} \int_{z_{i+1}}^{z_i} N(z')dz' \tag{6}$$

Where $F_{sun}$ is the solar irradiance at the top of atmosphere, $\tau_{sun,i}$ is the transmittance from space to level $i$, $\sigma_s$ the Rayleigh scattering cross-section, $P(\theta_i^{\uparrow\downarrow})$ are the upward and downward scattering phase functions calculated for layer $i$, and $\theta_{sat,i}$ is the satellite zenith angle in layer $i$, and $z_i$ is the height of level $i$. $N(z)$ is the number of particles per unit volume and height $z$. The factor of $4\pi$ normalises the phase function and the dependence on the sun-satellite azimuth is omitted here.


In the original RTTOV it was assumed the atmospheric layer was optically thin so that equal weight can be given to the radiance emitted from all points within the layer so the average value of the Planck function was used which is sufficient for clear sky calculations. For optically thick layers (e.g. with cloud) only the upper regions of

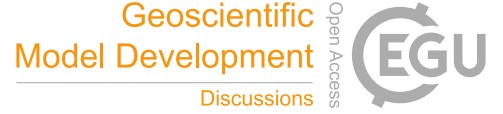

the layer give a significant contribution to the radiance. In this case the use of the average value of the Planck function would put too much weight on the radiance coming from the lower part of the layer. To improve the accuracy a parameterisation of the Planck function based on the linear in $\tau$ assumption that the source function throughout the layer is a linear function of the optical depth $\tau$ is used:

$$B[T(\tau)] = B_0 + (B_1 - B_0)\frac{\tau}{\tau_*} \tag{7}$$

where $B_0$ is the Planck function for the top of the layer, $B_1$ is the Planck function at the bottom of the layer and $\tau_*$ is the optical depth of the layer.

RTTOV can also estimate daytime Non-Local Thermodynamic Equilibrium (NLTE) effects in the $CO_2$ $v_3$ band
(around 4.3 μm). Here, local thermodynamic equilibrium breaks down due to the absorption of the incident solar radiation. The effect can add around 10 K to the measured brightness temperatures at midday. NLTE effects are introduced in the RTTOV calculations by adding a correction to the standard LTE radiances for affected channels (between 2200 cm$^{-1}$ to 2400 cm$^{-1}$) of high resolution IR sounders. The most recent NLTE correction is computed using a predictor-based regression scheme (Matricardi et al., 2017). The predictors consist of various
combinations of the solar zenith angle, the sensor zenith angle, and the average kinetic temperature in two broad layers above ~51 hPa. The regression has been trained using a database of accurate vibrational temperatures computed using the Granada NLTE population algorithm (Funke et al., 2012).

**3.4 Cloud, precipitation and aerosol affected radiance simulations**

RTTOV offers a number of approaches for simulating the radiative effect of cloud, precipitation and aerosols, each tailored to its own frequency domain as defined in Table 2. In the microwave and infrared, there are options to treat clouds as simple absorbers; such approaches are fast but their validity is limited mainly to water clouds. To accurately simulate the effect of liquid and frozen precipitation in the microwave, and cloud and aerosol in the visible and infrared, it is necessary to represent the effects of multiple scattering. Hence a number of more
sophisticated models are also available.

| Option in RTTOV | Microwave | Infrared | Visible |
|---|---|---|---|
| **Simple cloud (no scattering)** | | | |
| Grey optically thick cloud | No | Yes | Yes |
| Liquid water absorption | Yes (through normal RTTOV interface) | No | No |
| | | | |
| **Scattering solutions** | | | |
| Delta-Eddington | Yes (through RTTOV-SCATT interface) | No | No |





| Chou Scaling | No | Yes | No |
| Discrete Ordinates | No | Yes | Yes |

Table 2. Options in RTTOV for simulating the effects of cloud, precipitation and aerosols

### 3.4.1 Cloud and aerosol radiance simulations at infrared wavelengths

For the cloudy sky radiances there are two options, a simple uniform grey cloud assumption and a more complex scattering calculation for complex clouds. The 'simple' cloudy radiance model is given by:

$$L^{Cld}(\nu, \theta_{sat}) = \tau_{Cld}.B(\nu, T_{Cld}) + \int_{\tau_{Cld}}^{1} B(\nu, T)d\tau \qquad (8)$$

Where $\tau_{Cld}$ is the transmittance from the uniform cloud top to the top of atmosphere and $T_{Cld}$ is the cloud top temperature. This formulation can be used for simulating radiances over uniform grey cloud and has been used to extend the use of clear sky radiances in data assimilation to uniform cloud situations.

For more complex cloudy fields the parameterisation of multiple scattering in the IR, introduced in RTTOVv9, is designed to avoid solving the full scattering equation but instead to solve a modified version of Eq. 4 in which the absorption optical depth is replaced by an effective optical depth for the extinction. This approximation enables the efficiency of the RTTOV layer optical depth computation to be retained.

The RTTOV parameterisation of multiple scattering is based on the approach proposed by Chou et al. (1999) who originally developed a scheme to compute approximate fluxes in a scattering atmosphere. In the scheme by Chou et al. (1999), the radiative transfer equation is identical to the one with no scattering but, crucially, the optical depth for absorption, $\tau$, is replaced by an effective optical depth for extinction, $\tau_e$ :

$$\tau_e = \tau + b\tau_{sc} \qquad (9)$$

Where $\tau_{sc}$ is the optical depth for scattering and $b$ is the mean fraction of the radiation scattered in the upward direction for isotropic radiation incident from above (see Chou et al., 1999 for details). To derive their approximated form of the scattering transfer equation, Chou et al. (1999) have folded the effect of backscattering into a contribution to atmospheric emission and absorption. In addition, they have assumed that the diffuse radiance field is isotropic and can be approximated by the local Planck function.

To compute $b$ a knowledge of the phase function of the atmospheric particulates to be considered is needed which can be from aerosols, water droplets or ice crystals. For RTTOVv12 IR radiances in the presence of thirteen different types of aerosol components, five different types of water clouds and two options for ice cloud properties can be computed.

To solve the radiative transfer for an atmosphere partially covered by clouds, a stream method is used that divides the field of view into a number of horizontally homogeneous columns, each column containing either cloud-free layers or totally cloudy layers. Each column is assigned a fractional coverage and the number of columns is determined by the cloud overlapping assumption (maximum-random overlap) and the number of
layers the atmosphere is divided up into. The total radiance is then obtained as the sum of the radiances for the single columns weighted by the column fractional coverage.





For aerosols, the range and shape vary from quasi-spherical to highly irregular with a size typically less than 1 μm although particles as large as 10 to 20 μm have been observed. A database of optical properties assuming spherical particles has been generated using the microphysical properties assembled in the Optical Properties of Aerosols and Clouds (OPAC) software package (Hess et al., 1998) with some additional components added (old

and new volcanic ash and Asian dust). This database provides the microphysical properties (size distribution and refractive indices) for thirteen aerosol components (insoluble, water soluble, soot, two sea salt (accumulation mode and coarse mode), four mineral (nucleation, accumulation, coarse modes and transported), sulphated droplets and volcanic ash). Additionally a new volcanic ash and Asian dust components were provided in RTTOVv11. Some components are hydrophilic and so the properties are interpolated according to the local

relative humidity. For water clouds a cloud liquid water content must be provided in one or more atmospheric layers. It is then converted into a particle number concentration and the absorption, scattering and extinction optical depths can be derived from the normalized values of the optical parameters. RTTOV has parameters from OPAC for two stratus cloud types (stratus continental and stratus maritime) and 3 cumulus clouds (cumulus continental clean, cumulus continental polluted, and cumulus maritime) with the size distribution described by

the modified gamma distribution.

There are two ice cloud optical property parameterisations included in RTTOV for the VIS and IR wavelengths. The first uses the ice crystal properties dataset developed by (Baum et al., 2011) which is interpolated to obtain the scattering properties used in RTTOV (i.e. extinction coefficient ($\beta_{ext}$), single scattering albedo ($\omega_0$)) from the

input profile of ice crystal effective diameter. Users can explicitly provide ice effective diameter or can choose among four parameterisations in terms of ice water content and temperature (Ou and Liou, 1995; Wyser, 1998; Boudala et al., 2002; McFarquhar et al., 2003). The second scheme uses the methodology developed initially for the IR (Vidot et al., 2015) by using a large database of optical properties of ice clouds provided by Baran et al. (2014). It consists of 20662 particle size distributions using different *in situ* measured temperature (T) and

estimated ice water content (IWC) observations which simulates an ensemble of different ice cloud particle shapes which is expected to be more realistic than just assuming specific shapes as was done previously. It allows a direct parameterisation of the optical properties from the cloud temperature and the ice water content. For each pair of ice water content and temperature observations, the database contains the absorption and scattering coefficients, the asymmetry parameter (to compute the phase function), and also *b* (equation 10) used

for the Chou parameterisation at wavelengths between 0.2 and 19 microns. The formulation that has been implemented in RTTOV is given by the following equations:

$$\log_{10}[\beta_{ext}(\lambda, T, IWC)] = A_\beta(\lambda) + B_\beta(\lambda)T + C_\beta(\lambda)\log_{10}(IWC) + D_\beta(\lambda)T^2 +$$
$$E_\beta(\lambda)(\log_{10}(IWC))^2 + F_\beta(\lambda)T\log_{10}(IWC) \tag{11}$$

$$\overline{\omega_0}(\lambda, T, IWC) = A_{\overline{\omega}_0}(\lambda) + B_{\overline{\omega}_0}(\lambda)T + C_{\overline{\omega}_0}(\lambda)\log_{10}(IWC) \tag{12}$$

$$g(\lambda, T, IWC) = A_g(\lambda) + B_g(\lambda)T + C_g(\lambda)\log_{10}(IWC) \tag{13}$$

The parameterisation coefficients *A* to *F* of *βext, ω₀* and *g* the asymmetry parameter were calculated by using a non-linear least squares fitting procedure over the database, and are also functions of wavelength. Note the size distributions of the ice crystals assumed by the user (or their cloud model) should be consistent with the cloud

parameterisation scheme used in RTTOV. Users can input aerosol and/or cloud optical properties explicitly, so





they are not restricted to the optical properties assumed in RTTOV. The full details of this parameterisation of the scattering as implemented in RTTOV are documented in Matricardi (2005) and Saunders et al. (2017).

### 3.4.2 Cloudy and aerosol radiance simulations for solar radiation

For solar radiation, multiple scattering due to aerosols/clouds using the Discrete Ordinates Method or DOM
(Chandrasekhar, 1960) has been implemented in RTTOV for treating solar radiation and thermal emission. The choice of solver for the thermal emission and solar source terms can be selected independently: for thermal emission the choice is between the existing "Chou-scaling" parameterisation and DOM. For solar radiation only the DOM should be used.

The implementation of DOM is very similar to that in the DISORT model (Stamnes et al., 1988) such that the radiances from RTTOV agree to at least 4 significant figures with those from DISORT when equivalent inputs are used. The details of the DOM algorithm are given in Hocking (2015). There is one significant difference between the RTTOV and DISORT implementations of DOM: for solar simulations RTTOV takes the full phase functions as input and directly interpolates them at the scattering angle where required. In contrast DISORT
reconstructs the phase function from the full Legendre expansion. This is not a practical solution for some phase functions at VIS wavelengths which may require many thousands of Legendre terms in order to be accurately reconstructed. RTTOVv12 therefore only requires as many Legendre coefficients as there are discrete ordinates (or 'streams') in the calculation. The DOM implementation treats thermally emitted (IR) and solar radiation separately for reasons of efficiency. The surface is assumed to be Lambertian and in the IR the surface albedo is
calculated as (1-emissivity). For solar calculations the surface albedo is calculated as $\pi * BRDF$ and this value is capped at one to prevent unphysical albedo values being used. For each layer the absorption and scattering coefficients, the Legendre coefficients corresponding to the phase function and, for solar channels, the phase function itself have to be specified. The cloud and aerosol coefficient files include these properties for the aerosol and water cloud particle types defined in section 3.4. As for the IR the Baran et al. (2014) or Baum et al.
(2011) scattering properties datasets can be selected for ice cloud. The phase function is calculated from the asymmetry parameter following Baran et al. (2001) and the Legendre expansion of the phase function is calculated internally in RTTOV.

DOM is a solver for monochromatic radiances. However, RTTOV simulates radiances with a finite spectral
bandwidth and the standard RTTOV gaseous absorption optical depths are used as inputs to the DOM algorithm. The errors resulting specifically from applying DOM to polychromatic quantities were of the order of 1-2% in radiance for VIS/near-IR channels and the errors are dominated by variability in optical properties (especially the phase function) across the channel (Hocking, 2015). In the IR the errors are dominated by the variability of gas absorption across the channel and as the amount of scattering material in the atmosphere increases the errors
decrease because the optical properties of clouds/aerosols vary relatively slowly across the sensor channels and this begins to dominate over the gas absorption.

The DOM algorithm does not currently treat atmospheric Rayleigh scattering. It would be very expensive to compute as it would imply the presence of scattering particles in (almost) every layer. It is also the case that
currently the LBLRTM simulations used to train RTTOV include extinction due to Rayleigh scattering. If this



was disabled it would require an additional parameterisation of the Rayleigh extinction to be developed for clear-sky VIS/near-IR simulations. The existing Rayleigh single-scattering calculation is included as an "additive" effect alongside DOM so there is no interaction between the Rayleigh scattered radiation and the clouds/aerosols except for increased extinction by Rayleigh scattering (included in the gaseous optical depths used in DOM) and by clouds/aerosols (included in the Rayleigh single-scattering calculation). This leads to an underestimation of the top-of-atmosphere reflectances as the optical thickness of the scattering layers increases and as the wavelength decreases (Scheck, 2016a). Improvements to the treatment of Rayleigh scattering will be investigated for a future version of RTTOV. A faster scattering model at VIS wavelengths, MFASIS (Scheck et al., 2016b), is being developed for future versions of RTTOV which will allow VIS channel radiances to be used for real time data assimilation applications.

### 3.4.3 Scattering at microwave frequencies

Although scattering by hydrometeors (e.g. rain and snow) at MW frequencies is not included in the core RTTOV package there is a wrapper programme (RTTOV-SCATT, Bauer et al., 2006) that provides this capability outside of RTTOV. Full details of the initial formulation of the model are given in the RTTOV-8 Science and Validation Report (Saunders et al. 2006). RTTOV-SCATT is a multiple-scattering radiative transfer model which enables all-sky MW radiance assimilation in numerical weather prediction models and as for the core RTTOV code includes forward, tangent-linear, adjoint and Jacobian models. The all-sky brightness temperature is calculated, as in Eq. 3, as the combination of independent clear and cloudy columns weighted by an effective cloud fraction. The gaseous absorption component of both columns is computed by RTTOV. The scattering calculation in the cloudy column is based on the Delta-Eddington approximation (Joseph et al., 1976) so that only one angle (i.e. the observation angle) is needed and the anisotropic radiance field is decomposed into an isotropic and anisotropic component. Compared to reference doubling-adding simulations, this produces mean errors of less than 0.5 K at the targeted MW frequencies between 10 and 200 GHz, based on a dataset of 8290 model profiles located in tropical areas to ensure the presence of deep clouds and intense precipitation so that multiple scattering is maximized (Bauer et al., 2006).

The hydrometeor types assumed in RTTOV-SCATT are rain, snow, cloud liquid water and cloud ice. Tables of hydrometeor optical properties are pre-calculated for the required frequencies, temperatures and hydrometeor classes. As a function of hydrometeor water content, these give the bulk (i.e. integrated over an assumed particle size spectra) extinction coefficient, single scattering albedo and asymmetry parameter as required to perform the radiative transfer calculations. The optical properties are stored in sensor specific coefficient files, with calculations valid at the centre frequency of the channel, or if a channel is composed of multiple sidebands, the optical properties are an average of those at the centre frequencies of the sidebands. Cloud ice, cloud water and rain hydrometeors are represented by spheres, with their optical properties computed from Mie theory. Details of the underlying assumptions required for different hydrometeor types, i.e. particle permittivity as a function of frequency, temperature and water/ice content, as well as the particle size distributions, can be found in Bauer (2001). Since the original implementation of RTTOV-SCATT, the representation of snow hydrometeors has been changed to use discrete dipole simulations of non-spherical particles by Liu (2008). This more realistic representation of the complex 3D shapes of frozen particles has led to improved simulations of deep convective clouds, validated by comparing observations with simulations from the ECMWF model (Geer and Baordo,





2014). The code used to compute the optical property lookup tables is available as part of RTTOV, which gives the user the possibility to adjust the microphysical representations of hydrometeors if required.

An additional development since the original implementation of RTTOV-SCATT has been the revised
calculation of the effective cloud fraction in Eq. 3. The original approach used the maximum cloud fraction in the profile, which would be plausible choice for visible radiative transfer applications but generates excessive beamfilling at microwave frequencies. Geer et al. (2009) changed this to a hydrometeor-weighted average across the vertical profile of input cloud fraction; this decreased RMS errors by 40% compared to reference simulations.

Although the treatment of microwave scattering is highly simplified, including the treatment of sub-grid-scale cloud cover, the `one shape fits all' hydrometeor optical properties, and the use of essentially a two-stream solver for scattering, this model is used successfully at some centres for 'all-sky' MW radiance assimilation. It gives errors much smaller than the many other uncertainties involved, and critically, it is fast enough for operational use (e.g. Bauer et al., 2010; Geer et al., 2017).

**3.5 Simulating principal components of the infrared spectrum**

A major update to RTTOV at version 10 was the introduction of the capability of performing radiances from the IR spectrum in the form of principal components (PCs) which is a useful way to represent all the measurements from the advanced IR sounders such as IASI. The method used for the simulation of the PC scores for clear skies is described in Matricardi (2010) and referred to as PC-RTTOV. The dataset of atmospheric profiles used to train
the PC model consists of profiles generated using the operational suite of the ECMWF forecast model. It comprises 12500 vertical profiles of temperature, water vapour, ozone and ancillary information on surface properties over all surface types. Aerosols, trace gases and NLTE effects are all included in the latest version. The PC scores obtained from the eigenvectors of the covariance matrix of the simulated radiances are used in a linear regression scheme where they are expressed as a linear combination of profile dependent predictors. The
predictors consist of a selected number of polychromatic radiances computed using the standard RTTOV transmittance model described in section 3.2. The linear regression scheme can then be used to simulate PC scores and consequently reconstruct radiances for any input atmospheric profile.

The principal component option in RTTOV is much more computationally efficient for sensors like the advanced
IR sounders with many channels. The user can select the number of predictors used in the Principal Component (PC) score regression algorithm and the number of eigenvectors used in the reconstruction of the radiances. The different combinations of eigenvectors/predictors can provide a trade-off between more accurate but less computationally efficient, simulations. PC score regression coefficients are available based on 300, 400, 500, and 600 predictors for IASI whereas the number of eigenvectors can be up to 400. A detailed description of the PC-
RTTOV model and an assessment of its accuracy can be found in Matricardi (2010).

HT-FRTC is another model that calculates transmittance and radiance spectra using PCs (Havemann et al., 2009) and has been included as an optional module called from the RTTOV interface. The PCs cover the spectrum at very high spectral resolution, similar to that of conventional line-by-line models, so that individual spectral lines are resolved. This approach allows very fast calculations of spectra with line-by-line-like accuracy for clear





scenes. The PCs are derived from high spectral resolution data using a diverse set of atmospheres and surface conditions. Monochromatic calculations at these frequencies, are used to predict the PC scores (Havemann 2006). The liquid cloud optical properties are parameterised in terms of the droplet effective radius. The cirrus optical properties are based on Baran et al. (2014) and scattering by frozen and liquid water and aerosols are

approximated by modifications to the transmittances using the Chou approximation (Chou et al., 1999). An assessment of the HT-FRTC model for advanced IR sounder simulations is underway using the Met Office NWP model.

## 4 Performance of RTTOV compared with line-by-line models

There are several ways to assess the performance of a fast radiative transfer model: firstly to investigate the

accuracy of the fast model itself by comparing the primary outputs from RTTOV with the corresponding values computed using an accurate line-by-line model which is described in this section. Secondly to compare the computed radiances with observations where an underlying atmospheric state can be provided usually from an NWP model analysis or short range forecast for input to the fast model covered in the following section. The parameters assumed for the comparison between RTTOV and the line-by-line models for the VIS, IR and MW

regions are given in Table 3. The coefficients are computed from the 83 profile diverse set (Matricardi, 2008) and also comparisons were made for the MW channels on an independent 52 diverse profile set although the results were similar for both profile sets. RTTOVv12.1 is used throughout for these comparisons. The changes in brightness temperature due to updated spectroscopic parameters in the line-by-line models can change the computed values by several tenths of a degree but here we just document the accuracy of RTTOV reproducing

the line-by-line model optical depth values for VIS, IR and MW radiometers and advanced IR sounders.

Saunders et al. (1999) showed in their Figure 3 results for the AMSU MW sounder to demonstrate the performance of RTTOVv5 for MW radiometers and so a similar comparison for RTTOVv12 is given in Figure 1 which shows the differences for the AMSU channels for the 52 profile independent profile set. The upper

tropospheric water vapour channel of AMSU-B/MHS (channel 18) has the largest standard deviation with the line-by-line reference (0.04 K) but this is an order of magnitude below the instrument noise. The largest bias for channel 20 is only 0.002 K. The equivalent results shown in Saunders et al. (1999) show much higher standard



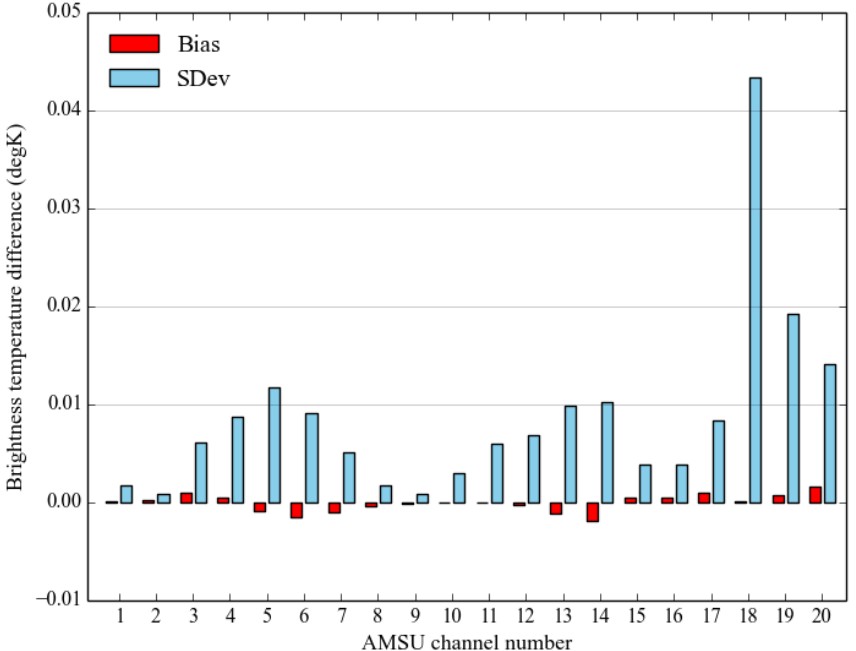

Figure 1: The mean differences and standard deviations for AMSU(1-15)/MHS(1-5) channels between the AMSUTRAN line-by-line model and RTTOVv12 for the 52 profiles independent set for viewing angles out to 63 deg using version 7 predictors.

| Parameters | RTTOV v12 Simulations |
|---|---|
| Number of layers for optical depth calculation | VIS/IR/MW 54 (0.005-1050hPa)<br>IASI 101 (0.005-1100hPa) |
| Input Profile sets | Dependent 83 Profiles<br>Independent 52 Profiles |
| **VIS/IR Transmittances** | |
| Spectroscopic data | LBLRTMv12.2/<br>AER 3.2, MTCKD2.5.2 |
| Surface emissivity assumed | 1.0 |
| Surface reflectance assumed | BRDF of $0.3/\pi$ |
| Optical depth predictors | Version 7 for HIRS & version 9 for IASI and VIS channels |
| **MW Transmittances** | |
| Spectroscopic data | Liebe et al., 1989 update/<br>Tretyakov 2005 |
| Surface emissivity assumed | 1.0 |
| Optical depth predictors | Version 7 |

Table 3. The parameters assumed for RTTOV v12.1 vs line-by-line model comparisons.

deviation values for all AMSU channels (e.g. channel 18 is 0.2K). The big improvement is mainly due to improved predictors for water vapour absorption described in Matricardi et al. (2001) introduced from





RTTOVv7 onwards (referred to as version 7 predictors), but also better more diverse profile datasets with more profiles (32 TIGR vs 83 ECMWF) and more levels (40 vs 54) have also helped.

For HIRS the differences from the line-by-line model are shown in Figure 2, this time for the 83 profile
dependent set though the results are similar for the independent profiles. The results are harder to compare with those shown in Figure 1 of Saunders et al. (1999) as the profile dataset they used then was much bigger. An unpublished report at the time showing standard deviations of the differences with a 32 profile dataset for RTTOVv5 shows the errors for the HIRS temperature sounding channels are only slightly improved in RTTOVv12 but the errors for the water vapour channels have been reduced by a factor of 2 using the version 7
predictors.

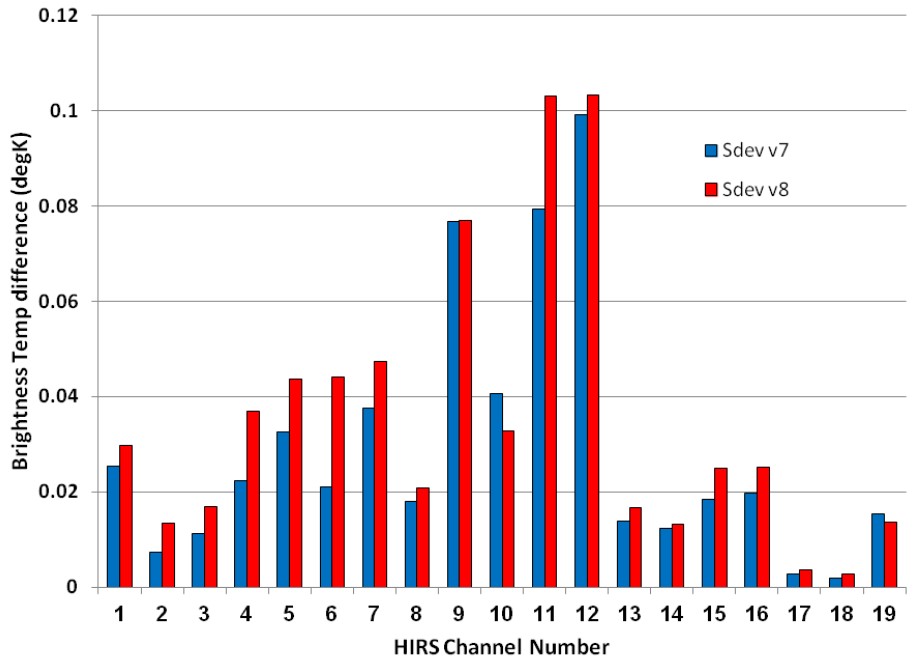

Figure 2: The mean standard deviation of the difference for METOP-A HIRS channels between the LBLRTM line by-line model and RTTOVv12 for the 83 profiles set for all nadir views up to 63 degrees using version 7 and 8 predictors.

Radiances from the advanced IR sounders are now a key part of the satellite observing system and RTTOV has
been developed to simulate these accurately (Matricardi, 2003; Matricardi, 2005). An example of simulations for the IASI sounder is shown in Figure 3 using both the older version 7 and the new version 9 predictors (Matricardi, 2008) which allow transmittances from a variety of trace gases to be computed in addition to water vapour and ozone (see section 3.2). In terms of the standard deviation of the differences the version 9 predictors are more accurate for the water vapour sensitive channels but the version 7 predictors are better for the
temperature sounding channels and window regions of the spectrum. However it has to be borne in mind that version 9 predictors have to explain the variability of trace gases in addition to temperature, water vapour and ozone whereas version 7 predictors do not. The differences seen are generally below the IASI instrument noise at all wavelengths. Version 9 predictors are mandatory if you want to include the simulation of traces gases which



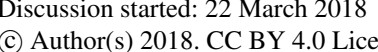


are not fixed in amount and also for simulations which include solar affected channels due to the larger zenith angles involved.

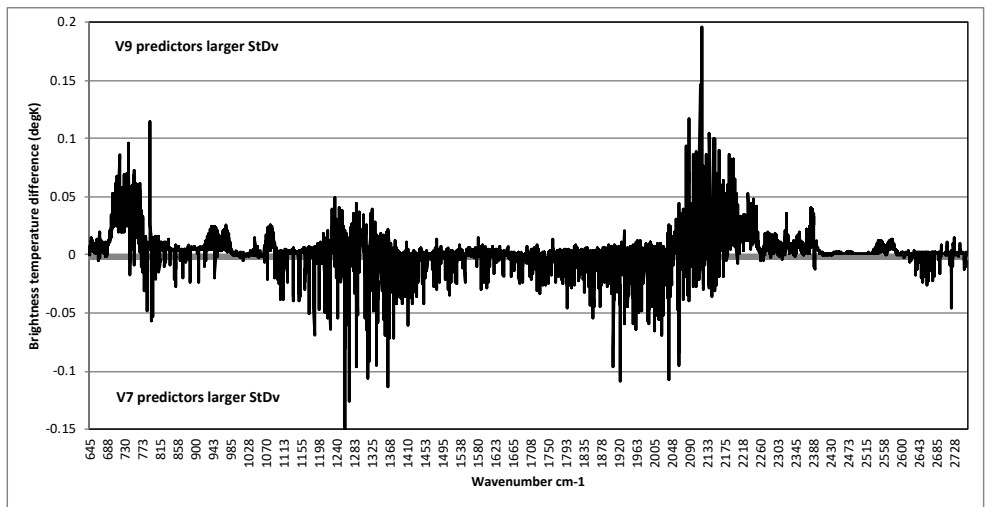

Figure 3: For selected IASI channels the difference of the standard deviations of the brightness temperature fit between the
LBLRTM line-by-line channel integrated optical depth model and RTTOVv12 computed optical depths for the 83 profiles dependent set for all views up to 63 degrees with version 7 and version 9 predictors. Version 7 predictors have smaller differences for positive values and version 9 predictors for negative values. Note the version 9 predictors are also computing the optical depths of the trace gases whereas for version 7 only water vapour and ozone are considered.

For simulations of imager radiances Figure 4 shows the comparison for the VIIRS VIS channels in terms of
reflectance for the range of satellite and solar zenith angles used to train the coefficients with a relative azimuth of 180 degrees. They include contributions from atmospheric Rayleigh scattering and surface reflection assuming a surface BRDF of $0.3/\pi$ located at the bottom level of the coefficient pressure profile (1050hPa for 54L coefficients). The 83 profile set of profiles were used here. The channels affected by water vapour are the ones with higher standard deviations but the differences are all small. Note that the 0.65 micron channel is for
the much wider band day-night channel. Similar differences are seen for the ABI VIS/NIR channels on GOES.

Another aspect of the RTTOV radiance computations is to evaluate the gradient of the radiances as a function of changes in the state vector as this is important for data assimilation applications where the adjoint or Jacobian is used. A comparison of Jacobians from a set of line-by-line and fast radiative transfer models was undertaken by
Garand et al. (2001) for HIRS and AMSU and by Saunders et al. (2007) for AIRS showing that RTTOVv6 did match the line-by-line models well in these cases. Computing Jacobians from line-by-line models is costly requiring a separate run for each level and parameter in the state vector where they are perturbed relative to the reference profile. Figure 5 shows the temperature Jacobian for a temperature sounding channel of AMSU-A (channel 6, 54.4 GHz) and a water vapour Jacobian for a channel of AMSU-B (channel 3, 183±1 GHz) for a
typical 50 level tropical standard atmosphere together with those computed from the MW line-by-line model, AMSUTRAN. The agreement between the line-by-line model and the equivalent RTTOV generated Jacobian is close.





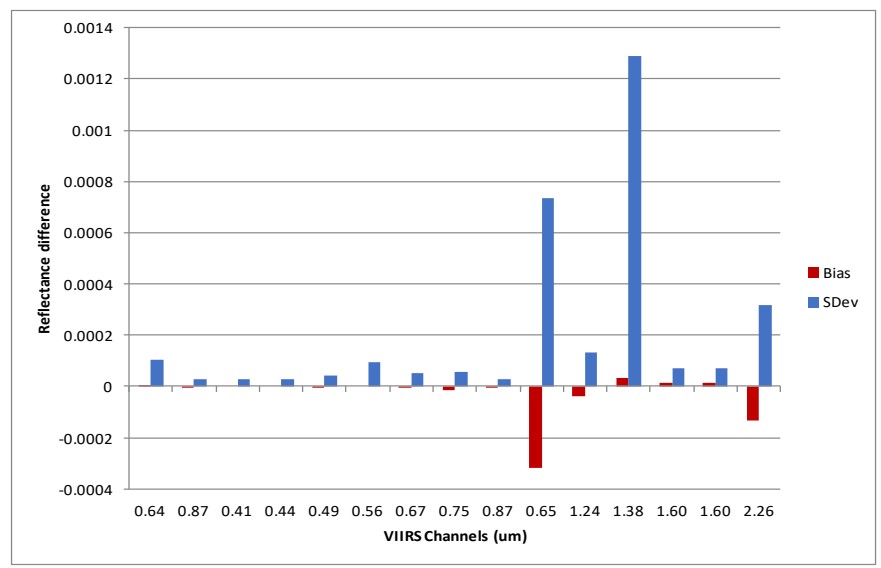

Figure 4. The mean difference for VIIRS visible/near IR channels between the HITRAN line-by-line model and RTTOVv12 for the 83 profiles set for all clear sky views with zenith angles up to 84 degrees using version 9 predictors.

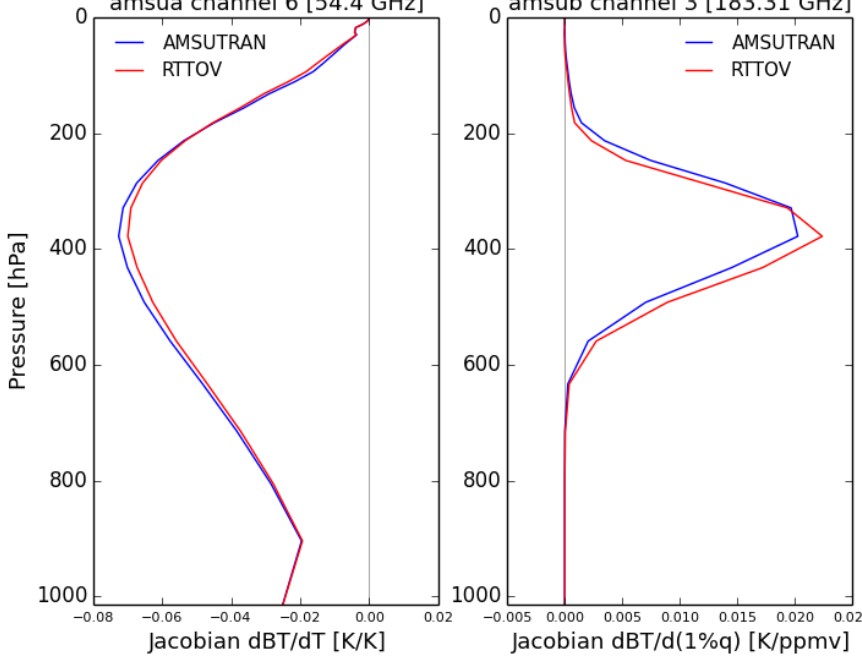

5    Figure 5. Temperature Jacobian for AMSU-A channel 5 (left) and water vapour Jacobian for AMSU-B channel 3 for the US standard tropical atmosphere. AMSUTRAN (blue) is from the line-by-line code and RTTOV (red) is using the RTTOV predictors to compute the transmittances for the Jacobians.





## 5 Comparisons with measurements

Another way to evaluate RTTOV is to compare the simulated radiances with real observations from a variety of sensors using a global NWP model to provide the atmospheric state coincident with the observation locations for input to RTTOV. This ensures a wide range of atmospheric conditions are sampled but biases due to instrument

calibration and NWP model errors are all included in the difference statistics. A reduction of the standard deviation of the differences however can be used as a measure of the improvement of the radiative transfer model performance if only the RT model in the system has changed. The observations compared with the model first guess profiles are from a 6 hr forecast. To demonstrate this, experiments have been run using the ECMWF model for a one month period 2 May - 2 June 2016 at reduced (TCo399) horizontal resolution and 137 vertical

levels with the model top pressure at 0.01 hPa. Version 7 optical depth predictors are used for all IR and MW observations, except for IASI, CrIS and AIRS where version 8 predictors were used which are different for the water vapour optical depth calculations. For HIRS only those instruments on MetOp-A and MetOp-B were considered but for AMSU-A instruments on 6 satellites (NOAA-15, NOAA-18, NOAA-19, MetOp-A, MetOp-B, Aqua) were included and for MHS instruments on NOAA-18, NOAA-19, MetOp-A and MetOp-B were used to

generate the statistics. Figure 6 shows the mean differences over the month for a selection of HIRS and AMSU sounding channels. The HIRS and AMSU-A temperature sounding channels have mean biases and standard deviations of differences all within 0.5 K. The water vapour channels of HIRS (channels 11 and 12) and AMSU (channels 38-40) have larger standard deviations (1-3 K) mainly due to the water vapour fields from the NWP model not being so accurate. HIRS channel 1 peaks high in the stratosphere and so NWP model uncertainties

again dominate the errors here.

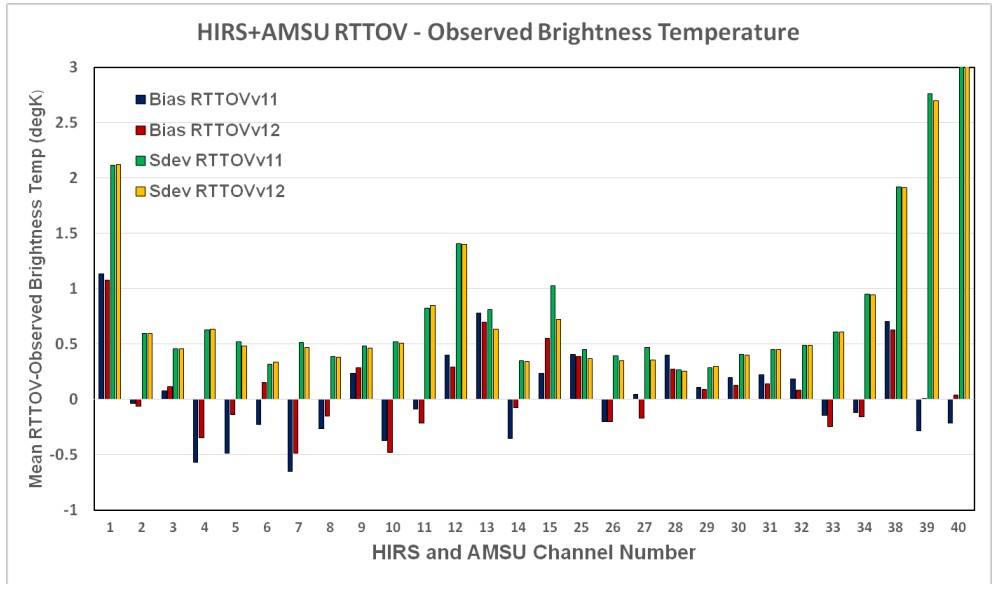

Figure 6. Mean global RTTOV minus observed statistics before bias correction for HIRS (channel 1-20) and AMSU-A
(channels 5-14 or 25-34 here) and MHS (channels 3-5 or 38-40 here) for the period 2 May - 2 June 2016 for RTTOV versions 11 and 12 in the ECMWF IFS NWP model.





Another factor for those channels which see the surface is the uncertainty in the surface emissivity assumed. The differences between the RTTOVv11 and RTTOVv12 computed radiances were in terms of bias slightly reduced for v12 but in terms of standard deviation the differences were in most cases negligible. The reduction in the bias is due to improved spectroscopy in this case. Figure 7 shows calculations for a selection of IASI channels across

the spectrum for RTTOVv11 and v12 with coefficients that are based on different versions of the line-by-line models (the main reason for the differences).

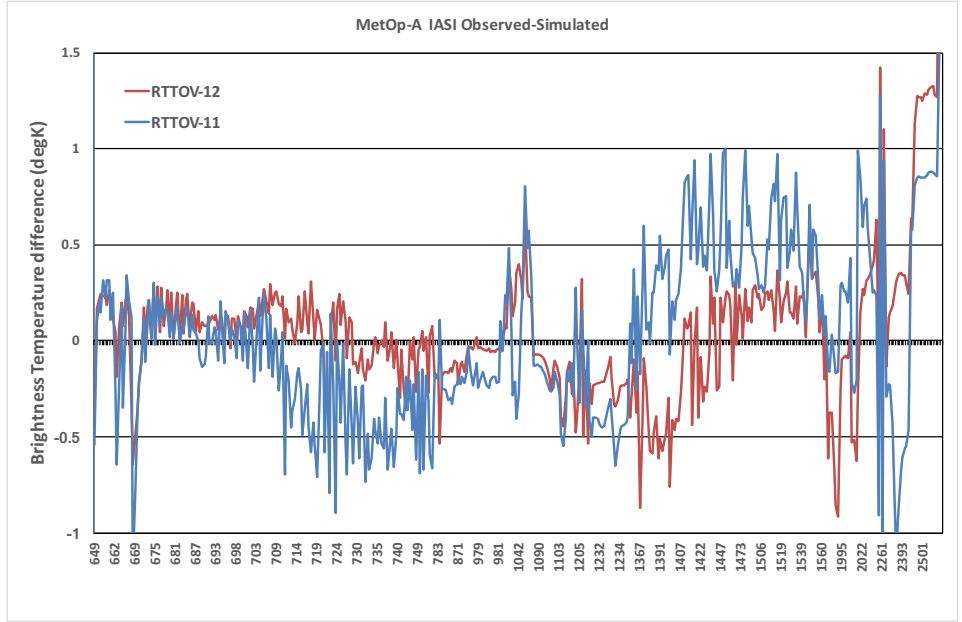

Figure 7. Mean global RTTOV minus observed statistics before bias correction for selected IASI channels for the period 2
May - 2 June 2016 for RTTOV versions 11 and 12 in the ECMWF IFS NWP model.

The biases were generally within ±1 degK and the RTTOVv12 values with updated spectroscopy reduced the bias with the model relative to RTTOVv11. The coefficients used with RTTOVv11 have been computed using the kCARTA line-by-line model while the newly released IR coefficient files with RTTOVv12 are based on LBLRTMv12.2. Other changes in the IR coefficient files include updated concentrations of $CO_2$ to current

values in the mixed gas transmissions and a different training set of diverse atmospheric profiles. The biases in Figure 7 are considerably larger than the comparisons of RTTOV with the line-by-line model (Figure 3) and are due to instrumental calibration biases and also biases in the NWP model temperature and humidity fields. The reduction in bias for RTTOVv12 shows that improvements in the underlying transmittances from the line-by-line models can significantly reduce the biases with measurements motivating further improvements in the

spectroscopic parameters. NWP centres apply a bias correction to remove these mean biases before assimilation.

## 6  Summary and Future Plans

The RTTOV fast radiative transfer model was first developed in the early 1990s and made available to researchers working with satellite sounding data for their physical retrievals and development of direct radiance assimilation in NWP models. Since that time the model development has been continuous under the auspices of





the EUMETSAT NWP SAF and its capabilities have greatly increased both in terms of wavelengths covered but also using updated spectroscopic parameters and improved diverse profiles. The computation of the layer optical depths has also been updated several times improving their accuracy and hence top of atmosphere radiances. RTTOV now has a user base which is truly worldwide with over 1000 registered users and is used at many
operational NWP centres for their radiance assimilation. The accuracy of the model at MW frequencies is shown to be much better than the instrument noise of the current MW radiometers. At IR wavelengths its accuracy is generally below the instrument noise of the current IR hyperspectral sounders but improvements for the water vapour affected channels are still needed for the clear sky radiative transfer. It is important to note that the largest differences between versions are often due to changes in the transmittances from the underlying line-by-
line models where spectroscopic parameters have been updated. Other improvements in the clear air radiative transfer include the representation of NLTE effects in the shortwave IR region during daylight and the inclusion of the Zeeman splitting of the oxygen lines around 60 GHz for upper stratosphere channels.

Concerning simulating cloudy radiances, the original version of RTTOV could only compute the radiance from a
fractional cover of uniform grey cloud layer at a defined level. The inclusion of cloud and hydrometeor scattering and absorption at each level and at all wavelengths within RTTOV has allowed cloudy, often referred to as 'all-sky', radiances to be computed for more complicated cloud regimes which match the observed radiances well (Aumann et al., 2018). This has proven useful for producing simulated satellite imagery from model fields to compare with the real imagery (Blackmore et al., 2014) and for developing an 'all-sky' radiance
assimilation system (Geer et al., 2017).

The representation of the land surface using emissivity models for RTTOV radiative transfer calculations was requested by users and this is an area continually under development with parameterised models used over the ocean and atlases available over land. Improved surface models are allowing more of the surface sensing
channels to be used actively for data assimilation and model validation.

To optimise the simulation of radiances from the advanced IR sounders RTTOV has included two options to compute principal components from the full IR spectrum, which is computationally much more efficient than computing the entire spectral range with thousands of channels. This capability is now being used to more fully
exploit the measurements from the hyperspectral sounders for data assimilation applications e.g. Matricardi and McNally (2014).

RTTOV will continue to be developed as part of the NWP SAF activities with new major versions planned every 3 years in response to user needs. In addition minor releases are made every year which fix bugs and provide
some limited upgrades which don't change the user interface. The RTTOV web pages (nwpsaf.eu) also provide bug fixes, coefficients for new instruments which can be downloaded by users and a variety of documentation for using RTTOV and associated studies.

The aspects of RTTOV which are planned to be improved in the coming years are:
• Assessing if the current layer optical depth prediction scheme can be further optimised in terms of accuracy, linearity and run time.





- Extending the range of clear sky simulations into the ultra-violet to allow the ozone sounding instruments to be simulated.
- Extending the MW simulations up to at least 700 GHz, including scattering, to allow new sensors planned to be modelled.
- To incorporate the MFASIS fast visible cloud scattering parameterisation into RTTOV.
- Adding 3D effects for cloud and hydrometeor scattering calculations
- Implementing a more efficient option for treating cloud overlap in VIS/IR cloud simulations.
- Developing the capability to simulate active MW sensors (e.g. radars) in RTTOV-SCATT.
- Extending PC-RTTOV and the interface to HT-FRTC to enable simulation of more situations (for example additional trace gases)

In addition to the above, improvements to the underlying spectroscopy at all wavelengths will be taken into account at each major RTTOV version and the coefficients regenerated using this updated spectroscopy.

## 7 Code and Data Availability

The latest version of the RTTOV model can be downloaded free of charge from the NWP SAF web site (nwpsaf.eu) once users have registered on the site to agree to the licence conditions. Updates to the code and coefficients for new instruments are also posted on the site. There is also a discussion forum for RTTOV (under the support tab) for users to share experiences with using the RTTOV model.

## 8 List of Acronyms

| AER | Atmospheric and Environmental Research |
|---|---|
| AIRS | Atmospheric InfraRed Sounder |
| AMSU | Advanced Microwave Sounding Unit |
| ATOVS | Advanced TIROS Operational Vertical Sounder |
| ATSR | Along Track Scanning Radiometer |
| AVHRR | Advanced Very High Resolution Radiometer |
| BRDF | Broadband Reflectance Distribution Function |
| CFMIP | Cloud Feedback Model Intercomparison Project |
| CNRM | Centre National de Recherches Météorologiques |
| COSP | CFMIP Observation Simulator Package |
| CrIS | Cross-track Infrared Sounder |
| CRTM | Community Radiative Transfer Model |
| DOM | Discrete Ordinates Method |
| ECMWF | European Centre for Medium-Range Weather Forecasts |
| EUMETSAT | European Organisation for the Exploitation of Meteorological Satellites |
| GMS | Geostationary Meteorological Satellite |
| GOES | Geostationary Operational Environmental Satellite |
| GUI | Graphical User Interface |



| HIRS | High resolution Infrared Sounder |
|---|---|
| HT-FRTC | Havemann-Taylor Fast Radiative Transfer Code |
| IASI | Infrared Atmospheric Sounding Interferometer |
| IR | Infrared |
| IWC | Ice Water Content |
| MFASIS | Method for FAst Satellite Image Synthesis |
| MHS | Microwave Humidity Sounder |
| MODIS | MODerate resolution Imaging Spectroradiometer |
| MSU | Microwave Sounding Unit |
| MT-CKD | Mlawer–Tobin_Clough– Kneizys–Davies continuum |
| MW | Microwave |
| NLTE | Non-Local Thermodynamic Equilibrium |
| NOAA | National Oceanic and Atmospheric Administration |
| NWP | Numerical Weather Prediction |
| OPAC | Optical Preperties of Aerosols and Clouds |
| PC | Principal Components |
| RTTOV | Radiative Transfer for TOVS |
| SAF | Satellite Application Facility |
| SSM/I | Special Sensor Microwave Imager |
| TELSEM | Tool to Estimate Land-Surface Emissivities at Microwave frequencies |
| TESSEM | Tool to Estimate Sea-Surface Emissivities at Microwave frequencies |
| TIGR | Thermodynamic Initial Guess Retrieval |
| TIROS | Television Infrared Observation Satellites |
| UWIREMIS | University of Wisconsin IR emissivity atlas |
| VIS | Visible |

**List of Figures**





## List of Tables

## Acknowledgements

Most of the RTTOV development is funded by the EUMETSAT NWP SAF over many years. Visiting scientists
funded by the NWP SAF have contributed over the years to various aspects of the development and their
contributions are gratefully acknowledged.

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
