# Peer review of "An update on the RTTOV fast radiative transfer model (currently at version 12)"

_Geoscientific Model Development, 2018_

## Referee Comment (RC1) · S. Boukabara (Referee) · 13 Apr 2018

This paper gives an overview of the widely-used RTTOV radiative transfer model. It is a nice description and summary of all the capabilities of the RTTOV system. From the formalism of the optical depth parameterization, to the description of the multiple scattering mechanism used in the simulation of measurements in cloudy conditions. While the paper represents a comprehensive description, it is nevertheless written in a relatively concise and very easy to follow fashion. It also gives a thorough overview of the history of the RTTOV, from inception to the latest version available on the NWP-SAF site. The paper does an excellent job reviewing the literature of existing (other) fast radiative transfer models (RTMs) and the assessment of their quality (both simulation and Jacobians). Speaking of quality, the paper covers the two aspects that are usually

employed to validate the assessment of fast RTMs: namely the (1) comparison to line-by-line models (more accurate but too slow to be applicable in real-time applications) and (2) comparison to real data from space-based observation systems. Which shows the significant improvement achieved over the last couple years. In summary, the paper is well written, it is comprehensive in its scope and the details it covers. It is envisioned that it will be not only a scientific reference, but also a sort of manual for all users of RTTOV for the coming years.

I look forward to other comments and discussions from the community.

Technical suggestion(s) : It is shown that standard deviation (when compared to line by line) is low, but that was computed with an emissivity background of 1. It is suspected that the standard deviation will be higher if the background is lower (higher contrast with the atmosphere), such as let's say 0.5 (more realistic over ocean). Can we have that assessed?

Also, it is good to have a low standard deviation and a low bias when tuning the absorption coefficients, but this does not exclude outliers that could happen, especially if the range of temperature, or other predictors, is outside the range of the training. Can this be assessed/shown?

―――――――――――――――――

---

## Referee Comment (RC2) · B. Ruston (Referee) · 21 May 2018

The manuscript presents the update of the RTTTOV model to version 12 and is a clear and concise summary of the history of the RTTOV model as well as many of the developments in fast radiative transfer for space-bourne assets. This manuscript can serve as a nice guide for the introduction of this scientific area for a newcomer to the field. The manuscript has no significant flaws in its language or descriptions, and makes judicious use of references to help provide the details which could not be included. I did see the comments from Dr. Boukabara, but for my part have no significant comments for modifications, and recommend the manuscript be published in its current form.

[Figure]

Minor suggestions: On page 1, when referencing TIROS TOVS radiometers could drop "old" before TIROS. On page 22 into 23, when discussing the aspects of RTTOV planned for the coming years, note that there is no mention of updating optical property tables for aerosols? Thought there was some momentum here, maybe is nothing concrete enough to be explicitly mentioned at this time?

---

## Author Response (AR1)

**Response to Reviewers Comments**

**Reviewer 1**

This paper gives an overview of the widely-used rttov radiative transfer model. it is a nice description and summary of all the capabilities of the rttov system. from the formalism of the optical depth parameterization, to the description of the multiple scattering mechanism used in the simulation of measurements in cloudy conditions. while the paper represents a comprehensive description, it is nevertheless written in a relatively concise and very easy to follow fashion. it also gives a thorough overview of the history of the rttov, from inception to the latest version available on the nwp-saf site. the paper does an excellent job reviewing the literature of existing (other) fast radiative transfer models (rtms) and the assessment of their quality (both simulation and jacobians). speaking of quality, the paper covers the two aspects that are usually employed to validate the assessment of fast rtms: namely the (1) comparison to line-by-line models (more accurate but too slow to be applicable in real-time applications) and (2) comparison to real data from space-based observation systems. which shows the significant improvement achieved over the last couple years. in summary, the paper is well written, it is comprehensive in its scope and the details it covers. it is envisioned that it will be not only a scientific reference, but also a sort of manual for all users of rttov for the coming years. i look forward to other comments and discussions from the community.

technical suggestion(s) : it is shown that standard deviation (when compared to line by line) is low, but that was computed with an emissivity background of 1. it is suspected that the standard deviation will be higher if the background is lower (higher contrast with the atmosphere), such as let's say 0.5 (more realistic over ocean). can we have that assessed?

also, it is good to have a low standard deviation and a low bias when tuning the absorption coefficients, but this does not exclude outliers that could happen, especially if the range of temperature, or other predictors, is outside the range of the training. can this be assessed/shown?

**Response to reviewer 1**

Thanks for your comments which are much appreciated. Regarding your technical suggestions:

(i) The plots shown in Figures 1 + 2 are assessing the ability of the fast model to reproduce the layer line-by-line transmittances that is all. In this case it was thought a surface emissivity of 1 would make it easier to assess the transmittances. We are not doing a full validation of RTTOV here that is done in the later figures. We agree that non unit emissivity will increase the st devs but that is not the purpose of the figure we think it best to keep it simple and just address the transmittance calculation here which is the core of RTTOV.

(ii) RTTOV raises flags if any of the profile variables are outside the range used in the training and it is up to the user whether he uses RTTOV in these cases. Experience has shown that if the variables are up to 10% outside the limits there is not a significant degradation in

the performance but beyond that there can be. I will add a sentence or two on this in the paper and if possible a reference to the assessment of this which has been done several times.

**Reviewer 2**

The manuscript presents the update of the RTTTOV model to version 12 and is a clear and concise summary of the history of the RTTOV model as well as many of the developments in fast radiative transfer for space-bourne assets. This manuscript can serve as a nice guide for the introduction of this scientific area for a newcomer to the field. The manuscript has no significant flaws in its language or descriptions, and makes judicious use of references to help provide the details which could not be included. I did see the comments from Dr. Boukabara, but for my part have no significant comments for modifications, and recommend the manuscript be published in its current form.

Minor suggestions: On page 1, when referencing TIROS TOVS radiometers could drop "old" before TIROS. On page 22 into 23, when discussing the aspects of RTTOV planned for the coming years, note that there is no mention of updating optical property tables for aerosols? Thought there was some momentum here, maybe is nothing concrete enough to be explicitly mentioned at this time?

**Response to reviewer 2**

Thanks for your comments.

Regarding your minor suggestions:

(i) Drop 'old' before TIROS - OK

(ii) You are right there are developments planned on improving the radiative properties of aerosols. I will add a bullet on this in the final version.

[revised manuscript text omitted]
(\nu, \theta_{sat}, \theta_{sun}) = (1 - N)L^{Clr}(\nu, \theta_{sat}, \theta_{sun}) + NL^{Cld}(\nu, \theta_{sat}, \theta_{sun}) \tag{3}$$

where $L^{Clr}(\nu, \theta_{sat}, \theta_{sun})$ and $L^{Cld}(\nu, \theta_{sat}, \theta_{sun})$ are the clear sky and overcast sky radiances at a frequency $\nu$ and zenith angle $\theta_{sat}$ and solar zenith $\theta_{sun}$. $N$ is the effective fractional cloud amount (i.e. the product of the fractional cloud amount and the cloud emissivity assuming it is grey body). The top of atmosphere clear sky radiance includes the emitted radiation from the surface and reflected downward radiation (emitted, solar and diffuse) and the emitted radiation from the atmosphere:

$$L^{Clr}(\nu, \theta_{sat}, \theta_{sun}) = \tau_s(\nu, \theta). \epsilon_s(\nu, \theta). B(\nu, T_s) + \int_{\tau_s}^{1} B(\nu, T)d\tau + (1 - \epsilon_s(\nu, \theta_{sat})) \tau_s^2(\nu, \theta_{sat}) \int_{\tau_s}^{1} \frac{B(\nu,T)}{\tau^2} d\tau +$$

[revised manuscript text omitted]

Reviewer 1

This paper gives an overview of the widely-used rttov radiative transfer model. it is a nice description and summary of all the capabilities of the rttov system. from the
5  formalism of the optical depth parameterization, to the description of the multiple scattering mechanism used in the simulation of measurements in cloudy conditions. while the paper represents a comprehensive description, it is nevertheless written in a relatively concise and very easy to follow fashion. it also gives a thorough overview of the history of the rttov, from inception to the latest version available on the nwp-saf site. the paper
10  does an excellent job reviewing the literature of existing (other) fast radiative transfer models (rtms) and the assessment of their quality (both simulation and jacobians). speaking of quality, the paper covers the two aspects that are usually employed to validate the assessment of fast rtms: namely the (1) comparison to line-by-line models (more accurate but too slow to be applicable in real-time applications) and (2)
15  comparison to real data from space-based observation systems. which shows the significant improvement achieved over the last couple years. in summary, the paper is well written, it is comprehensive in its scope and the details it covers. it is envisioned that it will be not only a scientific reference, but also a sort of manual for all users of rttov for the coming years. i look forward to other comments and discussions from the
20  community.

technical suggestion(s) : it is shown that standard deviation (when compared to line by line) is low, but that was computed with an emissivity background of 1. it is suspected that the standard deviation will be higher if the background is lower (higher contrast with the atmosphere), such as let's say 0.5 (more realistic over ocean). can we have that
25  assessed?

also, it is good to have a low standard deviation and a low bias when tuning the absorption coefficients, but this does not exclude outliers that could happen, especially if the range of temperature, or other predictors, is outside the range of the training. can this be assessed/shown?

30  Response to reviewer 1

Thanks for your comments which are much appreciated. Regarding your technical suggestions:

(i) The plots shown in Figures 1 + 2 are assessing the ability of the fast model to reproduce the layer line-by-line transmittances that is all. In this case it was thought a
35  surface emissivity of 1 would make it easier to assess the transmittances. We are not doing a full validation of RTTOV here that is done in the later figures. We agree that non unit emissivity will increase the st devs but that is not the purpose of the figure we think it best to keep it simple and just address the transmittance calculation here which is the core of RTTOV.

(ii) RTTOV raises flags if any of the profile variables are outside the range used in the training and it is up to the user whether he uses RTTOV in these cases. Experience has shown that if the variables are up to 10% outside the limits there is not a significant degradation in the performance but beyond that there can be. I will add a sentence or two on this in the paper unfortunately there is no reference to the assessment of this which has been done several times.

Reviewer 2

The manuscript presents the update of the RTTTOV model to version 12 and is a clear and concise summary of the history of the RTTOV model as well as many of the developments in fast radiative transfer for space-bourne assets. This manuscript can serve as a nice guide for the introduction of this scientific area for a newcomer to the field. The manuscript has no significant flaws in its language or descriptions, and makes judicious use of references to help provide the details which could not be included. I did see the comments from Dr. Boukabara, but for my part have no significant comments for modifications, and recommend the manuscript be published in its current form.

Minor suggestions: On page 1, when referencing TIROS TOVS radiometers could drop "old" before TIROS. On page 22 into 23, when discussing the aspects of RTTOV planned for the coming years, note that there is no mention of updating optical property tables for aerosols? Thought there was some momentum here, maybe is nothing concrete enough to be explicitly mentioned at this time?

Response to reviewer 2

Thanks for your comments.

Regarding your minor suggestions:

(i) Drop 'old' before TIROS - OK

(ii) You are right there are developments planned on improving the radiative properties of aerosols. I will add a bullet on this in the final version.